# Sampling for Bayesian Program Learning

**Kevin Ellis**
Brain and Cognitive Sciences
MIT
ellisk@mit.edu

**Armando Solar-Lezama**
CSAIL
MIT
asolar@csail.mit.edu

**Joshua B. Tenenbaum**
Brain and Cognitive Sciences
MIT
jbt@mit.edu

## Abstract

Towards learning programs from data, we introduce the problem of sampling programs from posterior distributions conditioned on that data. Within this setting, we propose an algorithm that uses a symbolic solver to efficiently sample programs. The proposal combines constraint-based program synthesis with sampling via random parity constraints. We give theoretical guarantees on how well the samples approximate the true posterior, and have empirical results showing the algorithm is efficient in practice, evaluating our approach on 22 program learning problems in the domains of text editing and computer-aided programming.

## 1 Introduction

Learning programs from examples is a central problem in artificial intelligence, and many recent approaches draw on techniques from machine learning. Connectionist approaches, like the Neural Turing Machine [1, 2] and symbolic approaches, like Hierarchical Bayesian Program Learning [3, 4, 5], couple a probabilistic learning framework with either gradient- or sampling-based search procedures. In this work, we consider the problem of Bayesian inference over program spaces. We combine solver-based program synthesis [6] and sampling via random projections [7], showing how to sample from posterior distributions over programs where the samples come from a distribution provably arbitrarily close to the true posterior. The new approach is implemented in a system called PROGRAMSAMPLE and evaluated on a set of program induction problems that include list and string manipulation routines.

### 1.1 Motivation and problem statement

Consider the problem of learning string edit programs, a well studied domain for programming by example. Often end users provide these examples and are unwilling to give more than one instance, which leaves the target program highly ambiguous. We model this ambiguity by *sampling* string edit programs, allowing us to learn from very few examples (Figure 1) and offer different plausible solutions. Our sampler also incorporates a description-length prior to bias us towards simpler programs.

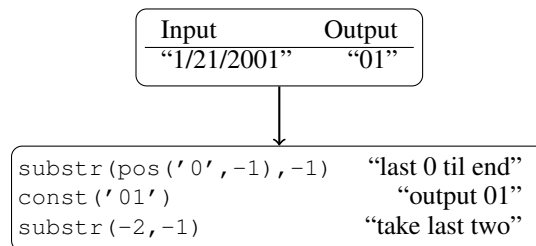

Figure 1: Learning string manipulation programs by example (top input/output pair). Our system receives data like that shown above and then sampled the programs shown below.

Another program learning domain comes from *computer-aided programming*, where the goal is to synthesize algorithms from either examples or formal specifications. This problem can be ill posed because many programs may satisfy the specification or examples. When this ambiguity arises, PROGRAMSAMPLE proposes multiple implementations with a bias towards shorter or simpler ones. The samples can also be used to efficiently approximate the posterior predictive distribution,

effectively integrating out the program. We show PROGRAMSAMPLE learning routines for counting and recursively sorting/reversing lists while modeling the uncertainty over the correct algorithm.

Because any model can be represented as a (probabilistic or deterministic) program, we need to carefully delimit the scope of this work. The programs we learn are a subset of those handled by constraint-based program synthesis tools. This means that the program is *finite* (bounded size, bounded runtime, bounded memory consumption), can be modeled in a constraint solver (like a SAT or SMT solver), and that the program's high-level structure is already given as a *sketch* [6], which can take the form of a recursive grammar over expressions. The sketch defines the search space and imparts prior knowledge. For example, we use one sketch when learning string edit programs and a different sketch when learning recursive list manipulation programs.

More formally, our sketch specifies a finite set of programs, $S$, as well as a measure of the programs' description length, which we write as $|x|$ for $x \in S$. This defines a prior ($\propto 2^{-|x|}$). For each program learning problem, we have a specification (such as consistency with input/output examples) and want to sample from $S$ conditioned upon the specification holding, giving the posterior over programs $\propto 2^{-|x|}\mathbb{1}[\textit{specification holds for } x]$. Throughout the rest of the paper, we write $p(\cdot)$ to mean this posterior distribution, and write $X$ to mean the set of all programs in $S$ consistent with the specification. So the problem is to sample from $p(x) = \frac{2^{-|x|}}{Z}$ where $Z = \sum_{x \in X} 2^{-|x|}$.

We can invoke a *solver*, which enumerates members of $X$, possibly subject to extra constraints, but without any guarantees on the order of enumeration. Throughout this work we use a SAT solver, and encode $x \in X$ in the values of $n$ Boolean decision variables. With a slight abuse of notation we will use $x$ to refer to both a member of $X$ and an assignment to those $n$ decision variables. An assignment to $j^{th}$ variable we write as $x_j$ for $1 \leq j \leq n$. Section 1.2 briskly summarizes the constraint-solving program synthesis approach.

## 1.2 Program synthesis by constraint solving

The constraint solving approach to program synthesis, pioneered in [6, 8], synthesizes programs by (1) modeling the space of programs as assignments to Boolean decision variables in a constraint satisfaction problem; (2) adding constraints to enforce consistency with a specification; (3) asking the solver to find any solution to the constraints; and (4) reinterpreting that solution as a program.

Figure 2 illustrates this approach for the toy problem of synthesizing programs in a language consisting of single-bit operators. Each program has one input ($i$ in Figure 2) which it transforms using `nand` gates. The grammar in Figure 2a is the sketch. If we inline the grammar, we can diagram the space of all programs as an AND/OR graph (Figure 2b), where $x_j$ are Boolean decision variables that control the program's structure. For each of the input/output examples (Figure 2d) we have constraints that model the program execution (Figure 2c) and enforce the desired output ($P_1$ taking value 1). After solving for a satisfying assignment to the $x_j$'s, we can read these off as a program (Figure 2e). In this work we measure the description length of a program $x$ as the number of bits required to specify its structure (so $|x|$ is a natural number).[1] PROGRAMSAMPLE further constrains unused bits to take a canonical form, such as all being zero. This causes the mapping between programs $x \in X$ and variable assignments $\{x_j\}_{j=1}^{N}$ to be one-to-one.

## 1.3 Algorithmic contribution

In the past decade different groups of researchers have concurrently developed solver-based techniques for (1) sampling of combinatorial spaces [9, 7, 10, 11] and (2) program synthesis [6, 8]. This work merges these two lines of research to attack the problem of program learning in a probabilistic setting. We use program synthesis tools to convert a program learning problem into a SAT formula. Then, rather than search for one program (formula solution), we augment the formula with random constraints that cause it to (approximately) sample the space of programs, effectively "upgrading" our SAT solver from a program synthesizer to a program sampler.

The groundbreaking algorithms in [9] gave the first scheme (XORSample) for sampling discrete spaces by adding random constraints to a constraint satisfaction problem. While one could use a tool like Sketch to reduce a program learning problem to SAT and then use an algorithm like XORSample,

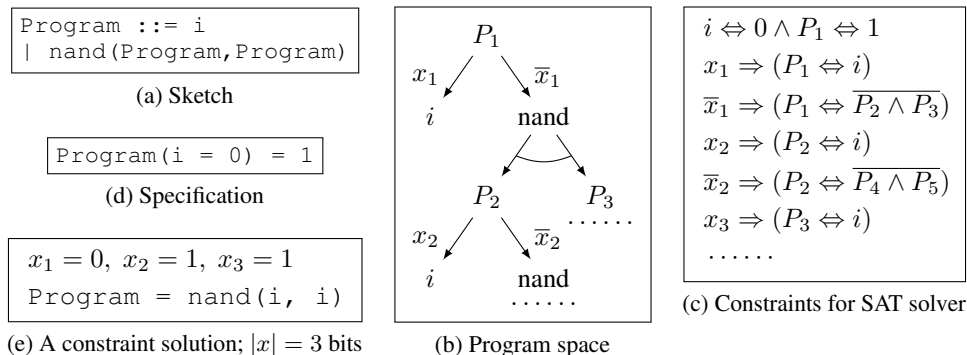

Figure 2: Synthesizing a program via sketching and constraint solving. `Typewriter font` refers to pieces of programs or sketches, while math font refers to pieces of a constraint satisfaction problem. The variable `i` is the program input.

PAWS, or WeightGen [9, 7, 10] to sample programs from a description length prior, doing so can be surprisingly inefficient[2]. The efficiency of these sampling algorithms depends critically on a quantity called the distribution's *tilt*, introduced in [10] as $\frac{\max_x p(x)}{\min_x p(x)}$. When there are a few very likely (short) programs and many extremely unlikely (long) programs, the posterior over programs becomes extremely tilted. Recent work has relied on upper bounding the tilt, often to around 20 [10]. For program sampling problems, we usually face very high tilt upwards of $2^{50}$. Our main algorithmic contribution is a new approach that extends these techniques to distributions with high tilt, such as those encountered in program induction.

## 2 The sampling algorithm

Given the distribution $p(\cdot)$ on the program space $X$, it is always possible to define a higher dimensional space $E$ (an *embedding*) and a mapping $F : E \to X$ such that sampling uniformly from $E$ and applying $F$ will give us approximately $p$-distributed samples [7]. But, when the tilt of $p(\cdot)$ becomes large, we found that such an approach is no longer practical.[3]

Our approach instead is to define an $F' : E \to X$ such that uniform samples on $E$ map to a distribution $q(\cdot)$ that is guaranteed to have low tilt, but whose KL divergence from $p(\cdot)$ is low. The discrepancy between the distributions $p(\cdot)$ and $q(\cdot)$ can be corrected through rejection sampling. Sampling uniformly from $E$ is itself not trivial, but a variety of techniques exist to approximate uniform sampling by adding random XOR constraints (random projections mod 2) to the set $E$, which is extensively studied in [9, 12, 10, 13, 11]. These techniques introduce approximation error that can be made arbitrarily small at the expense of lower efficiency. Figure 3 illustrates this process.

### 2.1 Getting high-quality samples

**Low-tilt approximation.** We introduce a parameter into the sampling algorithm, $d$, that parameterizes $q(\cdot)$. The parameter $d$ acts as a threshold, or cut-off, for the description length of a program; the distribution $q(\cdot)$ acts as though any program with description length exceeding $d$ can be encoded using $d$ bits. Concretely,

$$q(x) \propto \begin{cases} 2^{-|x|}, & \text{if } |x| \le d \\ 2^{-d}, & \text{otherwise} \end{cases} \tag{1}$$

If we could sample exactly from $q(\cdot)$, we could reject a sample $x$ with probability $1 - A(x)$ where $A$ is

$$A(x) \propto \begin{cases} 1, & \text{if } |x| \le d \\ 2^{-|x|+d}, & \text{otherwise} \end{cases} \tag{2}$$

and get exact samples from $p(\cdot)$, where the acceptance rate would approach 1 exponentially quickly in $d$. We have the following result; see supplement for proofs.

**Proposition 1.** *Let $x \in X$ be a sample from $q(\cdot)$. The probability of accepting $x$ is at least $\frac{1}{1+|X|2^{|x_*|-d}}$ where $x_* = \arg\min_x |x|$.*

The distribution $q(\cdot)$ is useful because we can guarantee that it has tilt bounded by $2^{d-|x_*|}$. Introducing the proposal $q(\cdot)$ effectively reifies the tilt, making it a parameter of the sampling algorithm, not the distribution over programs. We now show how to approximately sample from $q(\cdot)$ using a variant of the Embed and Project framework [7].

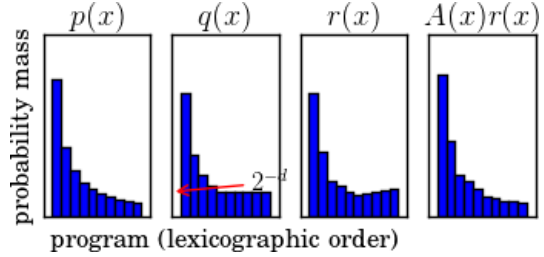

Figure 3: PROGRAMSAMPLE twice distorts the posterior distribution $p(\cdot)$. First it introduces a parameter $d$ that bounds the tilt; we correct for this by accepting samples w.p. $A(x)$. Second it samples from $q(\cdot)$ by drawing instead from $r(\cdot)$, where $KL(q||r)$ can be made arbitrarily small by appropriately setting another parameter, $K$. The distribution of samples is $A(x)r(x)$.

**The embedding.** The idea is to define a new set of programs, which we call $E$, such that short programs are included in the set much more often than long programs. Each program $x$ will be represented in $E$ by an amount proportional to $2^{-\min(|x|,d)}$, thus proportional to $q(x)$, such that sampling elements uniformly from $E$ samples according to $q(\cdot)$.

We embed $X$ within the larger set $E$ by introducing $d$ *auxiliary variables*, written $(y_1, \cdots, y_d)$, such that every element of $E$ is a tuple of an element of $x = (x_1, \cdots, x_n)$ and an assignment to $y = (y_1, \cdots, y_d)$:

$$E = \{(x,y) \; : \; x \in X, \bigwedge_{1 \leq j \leq d} |x| \geq j \Rightarrow y_j = 1\} \tag{3}$$

Suppose we sample $(x,y)$ uniformly from $E$. Then the probability of getting a particular $x \in X$ is proportional to $|\{(x',y) \in E : x' = x\}| = |\{y : |x| \geq j \Rightarrow y_j = 1\}| = 2^{\min(0,d-|x|)}$ which is proportional to $q(x)$. Notice that $|E|$ grows exponentially with $d$, and thus with the tilt of the $q(\cdot)$. This is the crux of the inefficiency of sampling from high-tilt distributions in these frameworks: these auxiliary variables combine with the random constraints to entangle otherwise independent Boolean decision variables, while also increasing the number of variables and clauses.

**The random projections.** We could sample exactly from $E$ by invoking the solver $|E|+1$ times to get every element of $E$, but in general it will have $O(|X|2^d)$ elements, which could be very large. Instead, we ask the solver for all the elements of $E$ consistent with $K$ random constraints such that (1) few elements of $E$ are likely to satisfy ("survive") the constraints, and (2) any element of $E$ is approximately equally likely to satisfy the constraints. We can then sample a survivor uniformly to get an approximate sample from $E$, an idea introduced in the XORSample' algorithm [9]. Although simple compared to recent approaches [10, 14, 15], it suffices for our theoretical and empirical results.

Our random constraints take the form of XOR, or parity constraints, which are random projections mod 2. Each constraint fixes the parity of a random subset of SAT variables in $x$ to either 1 or 0; thus any $x$ survives a constraint with probability $\frac{1}{2}$. A useful feature of random parity constraints is that whether an assignment to the SAT variables survives is independent of whether another, different assignment survives, which has been exploited to create a variety of approximate sampling algorithms [9, 12, 10, 13, 11].

Then the $K$ constraints are of the form $h\left(\begin{smallmatrix}x\\y\end{smallmatrix}\right) \overset{2}{\equiv} b$ where $h$ is a $K \times (d+n)$ binary matrix and $b$ is a $K$-dimensional binary vector. If no solutions satisfy the $K$ constraints then the sampling attempt is rejected. These samples are close to uniform in the following sense:

**Proposition 2.** *The probability of sampling $(x,y)$ is at least $\frac{1}{|E|} \times \frac{1}{1+2^K/|E|}$ and the probability of getting any sample at all is at least $1 - 2^K/|E|$.*

So we get approximate samples from $E$ as long as $|E|2^{-K}$ is not small. In reference to Figure 3, we call the distribution of these samples $r(x) = \sum_y r(x,y)$. Schemes more sophisticated than XORSample', like [7], also guarantee upper bounds on sampling probability, but we found that these

---

**Algorithm 1** PROGRAMSAMPLE

---

**Input:** Program space $X$, number of samples $N$, failure probability $\delta$, parameters $\Delta > 0, \gamma > 0$
**Output:** $N$ samples
Set $|x_*| = \min_{x \in X} |x|$
Set $B_X = \text{ApproximateUpperBoundModelCount}(X, \delta/2)$
Set $d = \lceil \gamma + \log B_X + |x_*| \rceil$
Define $E = \{(x, y) \ : \ x \in X, \bigwedge_{1 \leq j \leq d} |x| \geq j \implies y_j = 1\}$
Set $B_E = \text{ApproximateLowerBoundModelCount}(E, \delta/2)$
Set $K = \lfloor \log B_E - \Delta \rfloor$
Initialize samples $= [\,]$
**repeat**
    Sample $h$ uniformly from $\{0, 1\}^{(d+n) \times K}$
    Sample $b$ uniformly from $\{0, 1\}^K$
    Enumerate $\mathcal{S} = \{(x, y) \text{ where } h(x, y) = b \wedge x \in X\}$
    **if** $|\mathcal{S}| > 0$ **then**
        Sample $(x, y)$ uniformly from $\mathcal{S}$
        **if** $\text{Uniform}(0, 1) < 2^{d - |x|}$ **then**
            samples $=$ samples $+ [x]$
        **end if**
    **end if**
**until** $|\text{samples}| = N$
**return** samples

---

were unnecessary for our main result, which is that the KL between $p(\cdot)$ and $A(x)r(x)$ goes to zero exponentially quickly in a new quantity we call $\Delta$:

**Proposition 3.** *Write $Ar(x)$ to mean the distribution proportional to $A(x)r(x)$. Then $D(p||Ar) < \log\left(1 + \frac{1 + 2^{-\gamma}}{1 + 2^{\Delta}}\right)$ where $\Delta = \log|E| - K$ and $\gamma = d - \log|X| - |x_*|$.*

So we can approximate the true distribution $p(\cdot)$ arbitrarily well, but at the expense of either more calls to the solver (increasing $\Delta$) or a larger embedding (increasing $\gamma$; our main algorithmic contribution). See supplement for theoretical and empirical analyses of this accuracy/runtime trade-off.

Proposition 3 requires knowing $\min_x |x|$ to set $K$ and $d$. We compute $\min_x |x|$ using the iterative minimization routine in [16]; in practice this is very efficient for finite program spaces. We also need to calculate $|X|$ and $|E|$, which are model counts that are in general difficult to compute exactly. However, many approximate model counting schemes exist, which provide upper and lower bounds that hold with arbitrarily high probability. We use Hybrid-MBound [13] to upper bound $|X|$ and lower bound $|E|$ that each individually hold with probability at least $1 - \delta/2$, thus giving lower bounds on both the $\gamma$ and $\Delta$ parameters of Proposition 3 with probability at least $1 - \delta$ and thus an upper bound on the KL divergence. Algorithm 1 puts these ideas together.

## 3 Experimental results

We evaluated PROGRAMSAMPLE on program learning problems in a text editing domain and a list manipulation domain. For each domain, we wrote down a sketch and produced SAT formulas using the tool in [6], specifying a large but finite set of possible programs. This implicitly defined a description-length prior, where $|x|$ is the number of bits required to specify $x$ in the SAT encoding. We used CryptoMiniSAT [17], which can efficiently handle parity constraints.

### 3.1 Learning Text Edit Scripts

We applied our program sampling algorithm to a suite of programming by demonstration problems within a text editing domain. Here, the challenge is to learn a small text editing program from very few examples and apply that program to held out inputs. This problem is timely, given the widespread use of the FlashFill program synthesis tool, which now ships by default in Microsoft Excel [18] and can learn sophisticated edit operations in real time from examples. We modeled a subset of

the FlashFill language; our goal here is not to compete with FlashFill, which is cleverly engineered for its specific domain, but to study the behavior of our more general-purpose program learner in a real-world task. To impart domain knowledge, we used a sketch equivalent to Figure 4.

Because FlashFill's training set is not yet public, we drew text editing problems from [19] and adapted them to our subset of FlashFill, giving 19 problems, each with 5 training examples. The supplement contains these text edit problems.

We are interested both in the ability of the learner to generalize and in PROGRAMSAMPLE's ability to generate samples quickly. Table 1 shows the average time per sampling attempt using PROGRAMSAMPLE, which is on the order of a minute. These text edit problems come from distributions with extremely high tilt:

```
Program   ::= Term | Program + Term
Term      ::= String | substr(Pos,Pos)
Pos       ::= Number
            | pos(String,String,Number)
Number    ::= 0 | 1 | 2 | ...
            | -1 | -2 | ...
String    ::= Character
            | Character + String
Character ::= a | b | c | ...
```

Figure 4: The sketch (program space) for learning text edit scripts

often the smallest program is only tens of bits long, but the program space contains (implausible) solutions with over 100 bits. By putting $d$ to $|x_*| - n$ we eliminate the tilt correction and recover a variant of the approaches in [7]. This baseline does not produce any samples for any of our text edit problems in under an hour.[4] Other baselines also failed to produce samples in a reasonable amount of time (see supplement). For example, pure rejection sampling (drawing from the prior) is also infeasible, with consistent programs having prior probability $\leq 2^{-50}$ in some cases.

The learner generalizes to unseen examples, as Figure 5 shows. We evaluated the performance of the learner on held out test examples while varying training set size, and compare with baselines that either (1) enumerate programs in the arbitrary order provided by the underlying solver, or (2) takes the most likely program under $p(x)$ (MDL learner). The posterior is sharply peaked, with most samples being from the MAP solution, and so our learner does about as well as the MDL learner. However, sampling offers an (approximate) predictive posterior over predictions on the held out examples; in a real world scenario, one would offer the top $C$ predictions to the user and let them choose, much like how spelling correction works. This procedure allows us to offer the correct predictions more often than the MDL learner (Figure 6), because we correctly handle ambiguous problems like in Figure 1. We see this as a primary strength of the sampling approach to Bayesian program learning: when learning from one or a few examples, a point estimate of the posterior can often miss the mark.

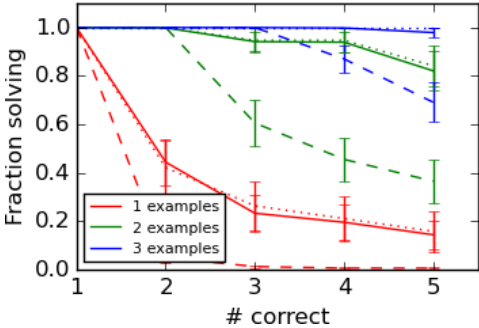

Figure 5: Generalization when learning text edit operations by example. Results averaged across 19 problems. Solid: 100 samples from PROGRAMSAMPLE . Dashed: enumerating 100 programs. Dotted: MDL learner. Test cases past 1 (respectively 2,3) examples are held out when trained on 1 (respectively 2,3) examples.

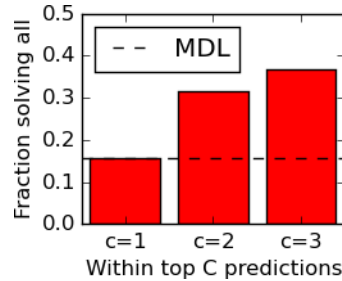

Figure 6: Comparing the MDL learner (dashed black line) to program sampling when doing one-shot learning. We count a problem as "solved" if the correct joint prediction to the test cases is in the top $C$ most frequent samples.

Table 1: Average solver time to generate a sample measured in seconds. See Figure 9 and 5 for training set sizes. $n \approx 180, \ 65$ for text edit, list manipulation domains, respectively. w/o tilt correction, sampling text edit & count takes $> 1$ hour.

|        | Large set | Medium set | Small set |
|--------|-----------|------------|-----------|
| text edit | $49\pm3$ | $21\pm1$ | $84\pm3$ |
| sort | $1549\pm155$ | $905\pm58$ | $463\pm65$ |
| reverse | $326\pm42$ | $141\pm18$ | $39\pm3$ |
| count | $\leq 1$ | $\leq 1$ | $\leq 1$ |

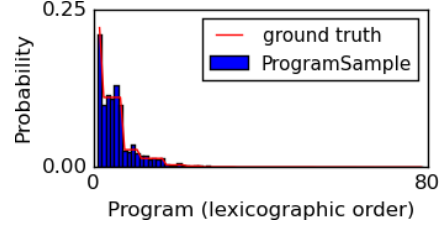

Figure 7: Sampling frequency vs. ground truth probability on a counting task with $\Delta = 3$ and $\gamma = 4$.

## 3.2 Learning list manipulation algorithms

One goal of program synthesis is *computer-aided programming* [6], which is the automatic generation of executable code from either declarative specifications or examples of desired behavior. Systems with this goal have been successfully applied to, for example, synthesizing intricate bitvector routines from specifications [18]. However, when learning from examples, there is often uncertainty over the correct program. While past approaches have handled this uncertainty within an optimization framework (see [20, 21, 16]), we show that PROGRAMSAMPLE can *sample* algorithms.

We take as our goal to learn recursive routines for sorting, reversing, and counting list elements from input/output examples, particularly in the ambiguous, unconstrained regime of few examples. We used a sketch with a set of basis primitives capable of representing a range of list manipulation routines equivalent to Figure 8.

A description-length prior that penalizes longer programs allowed learning of recursive list manipulation routines (from production `Program`) and a non-recursive count routine (from production `Int`); see Figure 9, which shows average accuracy on held out test data when trained on variable numbers of short randomly generated lists. With the large training set (5–11 examples) PROGRAMSAMPLE recovers a correct implementation, and with less data it recovers a distribution over programs that functions as a probabilistic algorithm despite being composed of only deterministic programs.

For some of these tasks the number of consistent programs is small enough that we can enumerate all of them, allowing us to compare our sampler with ground-truth probabilities. Figure 7 shows this comparison for a counting problem with 80 consistent programs, showing empirically that the tilt correction and random constraints do not significantly perturb the distribution.

Table 1 shows the average solver time per sample. Generating recursive routines like sorting and reversing is much more costly than generating the nonrecursive counting routine. The constraint-based approach propositionalizes higher-order constructs like recursion, and so reasoning about them is much more costly. Yet counting problems are highly tilted due to count's short implementation, which makes them intractable without our tilt correction.

```
Program ::=
  (if Bool List
    (append RecursiveList
            RecursiveList
            RecursiveList))
Bool ::= (<= Int) | (>= Int)
Int  ::= 0 | (1+ Int) | (1- Int)
       | (length List) | (head List)
List ::= nil | (filter Bool List)
       | X | (tail List) | (list Int)
RecursiveList ::= List
                | (recurse List)
```

Figure 8: The sketch (program space) for learning list manipulation routines; X is program input

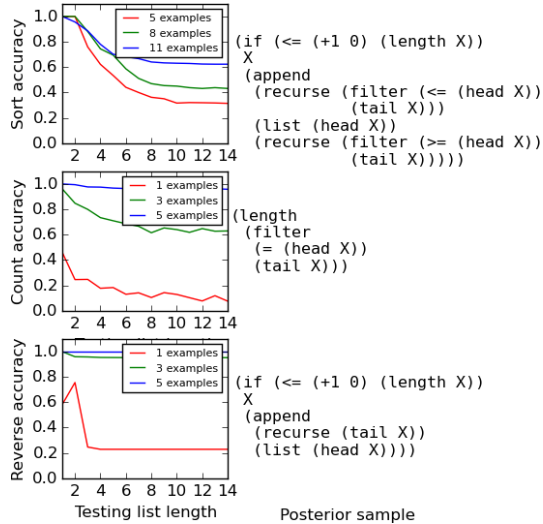

Figure 9: Learning to manipulate lists. Trained on lists of length $\leq 3$; tested on lists of length $\leq 14$.

# 4 Discussion

## 4.1 Related work

There is a vast literature on program learning in the AI and machine learning communities. Many employ a (possibly stochastic) heuristic search over structures using genetic programming [22] or MCMC [23]. These approaches often find good programs and can discover more high-level structure than our approach. However, they are prone to getting trapped in local minima and, when used as a sampler, lack theoretical guarantees. Other work has addressed learning priors over programs in a multitask setting [4, 5]. We see our work as particularly complementary to these methods: while they focus on learning the structure of the hypothesis space, we focus on efficiently sampling an already given hypothesis space (the sketch). Several recent proposals for recurrent deep networks can learn algorithms [2, 1]. We see our system working in a different regime, where we want to quickly learn an algorithm from a small number of examples or an ambiguous specification.

The program synthesis community has several recently proposed learners that work in an optimization framework [20, 21, 16]. By computing a posterior over programs, we can more effectively represent uncertainty, particularly in the small data limit, but at the cost of more computation.

PROGRAMSAMPLE borrows heavily from a line of work started in [9, 13] on sampling of combinatorial spaces using random XOR constraints. An exciting new approach is to use *sparse* XOR constraints [14, 15], which might sample more efficiently from our embedding of the program space.

## 4.2 Limitations of the approach

Constraint-based synthesis methods tend to excel in domains where the program structure is restricted by a sketch [6] and where much of the program's description length can be easily computed from the program text. For example, PROGRAMSAMPLE can synthesize text editing programs that are almost 60 bits long in a couple seconds, but spends 10 minutes synthesizing a recursive sorting routine that is shorter but where the program structure is less restricted. Constraint-based methods also require the entire problem to be represented symbolically, so they have trouble when the function to be synthesized involves difficult to analyze building blocks such as numerical routines. For such problems, stochastic search methods [23, 22] can be more effective because they only need to run the functions under consideration. Finally, past work shows empirically that these methods scale poorly with data set size, although this can be mitigated by considering data incrementally [21, 20].

The requirement of producing representative samples imposes additional overhead on our approach, so scalability can more limited than for standard symbolic techniques on some problems. For example, our method requires 1 MAP inference query, and 2 queries to an approximate model counter. These serve to "calibrate" the sampler, and its cost can be amortized because they only has to be invoked once in order to generate an arbitrary number of iid samples. Approximate model counters like MBound [13] have complexity comparable with that of generating a sample, but the complexity can depend on the number of solutions. Thus, for good performance, PROGRAMSAMPLE requires that there not be too many programs consistent with the data—the largest spaces considered in our experiments had $\leq 10^7$ programs. This limitation, together with the general performance characteristics of symbolic techniques, means that the approach will work best for "needle in a haystack" problems, where the space of possible programs is large but restricted in its structure, and where only a small fraction of the programs satisfy the constraints.

## 4.3 Future work

This work could naturally extend to other domains that involve inducing latent symbolic structure from small amounts of data, such as semantic parsing to logical forms [24], synthesizing motor programs [3], or learning relational theories [25]. These applications have some component of transfer learning, and building efficient program learners that can transfer inductive biases across tasks is a prime target for future research.

**Acknowledgments**

We are grateful for feedback from Adam Smith, Kuldeep Meel, and our anonymous reviewers. Work supported by NSF-1161775 and AFOSR award FA9550-16-1-0012.

## Footnotes

[1]This is equivalent to the assumption that $x$ is drawn from a probabilistic grammar specified by the sketch

[2]In many cases, slower than rejection sampling or enumerating all of the programs

[3][10] take a qualitatively different approach from [7] not based on an embedding, but which still becomes prohibitively expensive in the high-tilt regime.

[4]Approximate model counting of $E$ was also intractable in this regime, so we used the lower bound $|E| \geq 2^{d-|x_*|} + |X| - 1$

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
