[Supplementary Material]

# Supplement to: Sampling for Bayesian Program Learning

**Kevin Ellis**
Brain and Cognitive Sciences
MIT
ellisk@mit.edu

**Armando Solar-Lezama**
CSAIL
MIT
asolar@csail.mit.edu

**Joshua B. Tenenbaum**
Brain and Cognitive Sciences
MIT
jbt@mit.edu

## 1 Proofs

Due to space considerations we have included the proofs of our theoretical results here; see original paper for context.

**Proposition 1.** *Let* $x \in X$ *be a sample from* $q(\cdot)$. *The probability of accepting* $x$ *is at least* $\frac{1}{1+|X|2^{|x_*|-d}}$ *where* $x_* = \arg\min_x |x|$.

*Proof.* The probability of acceptance is $\sum_x q(x)A(x)$, or

$$\sum_x \frac{2^{-|x|}}{\sum_{x'} q(x')} = \frac{Z}{Z + \sum_{|x|>d}(2^{-d} - 2^{-|x|})} > \frac{1}{1+|X|2^{-d}/Z} > \frac{1}{1+|X|2^{|x_*|-d}}. \quad (1)$$

$\square$

**Proposition 2.** *The probability of sampling* $(x,y)$ *is at least* $\frac{1}{|E|} \times \frac{1}{1+2^K/|E|}$ *and the probability of getting any sample at all is at least* $1 - 2^K/|E|$.

*Proof.* The probability of sampling $(x,y)$, given that $(x,y)$ survives the $K$ constraints, is $\frac{1}{mc}$, where mc is the model count (# of survivors). The probability of $(x,y)$ surviving the $K$ constraints is $2^{-K}$ and is independent of whether any other element of $E$ survives the constraints [1]. So the probability of sampling $(x,y)$ is

$$2^{-K} \sum_{i=1}^{|E|} \mathbb{P}\left[mc = i | (x,y) \text{ survives}\right] \frac{1}{i} \quad (2)$$

$$= 2^{-K} \mathbb{E}\left[\frac{1}{mc} | (x,y) \text{ survives}\right] \quad (3)$$

$$> 2^{-K} \frac{1}{\mathbb{E}[mc|(x,y) \text{ survives}]}, \text{ Jensen's inequality} \quad (4)$$

$$= 2^{-K} \frac{1}{1 + (|E| - 1)2^{-K}}, \text{ pairwise independence} \quad (5)$$

$$> \frac{1}{|E|} \times \frac{1}{1 + 2^K/|E|}. \quad (6)$$

We fail to get a sample if mc $= 0$. We bound the probability of this event using Chebyshev's inequality: $\mathbb{E}[mc] = |E|2^{-K} > \text{Var}(mc)$, so

$$\mathbb{P}[mc = 0] \leq \mathbb{P}[|mc - \mathbb{E}[mc]| \geq \mathbb{E}[mc]] \quad (7)$$

$$\leq \frac{\text{Var}(mc)}{\mathbb{E}[mc]^2} < 1/\mathbb{E}[mc] = 2^K/|E|. \quad (8)$$

$\square$

**Proposition 3.** *Write $Ar(x)$ to mean the distribution proportional to $A(x)r(x)$. Then $D(p||Ar) < \log\left(1 + \frac{1+2^{-\gamma}}{1+2^{\Delta}}\right)$ where $\Delta = \log|E| - K$ and $\gamma = d - \log|X| - |x_*|$.*

*Proof.* Define $c = \frac{1}{1+2^K/|E|}$. As $p(x) \propto A(x)q(x)$,

$$D(p||Ar) = \sum_x p(x) \log \frac{p(x) \sum_y A(y)r(y)}{A(x)r(x)} \tag{9}$$

$$= \sum_x p(x) \log \frac{A(x)q(x)}{\sum_y A(y)q(y)} \frac{\sum_y A(y)r(y)}{A(x)r(x)} \tag{10}$$

$$= \log \frac{\sum_x A(x)r(x)}{\sum_x A(x)q(x)} + \sum_x p(x) \log \frac{q(x)}{r(x)} \tag{11}$$

$$< \log \frac{\sum_x A(x)r(x)}{\sum_x A(x)q(x)} - \log c \tag{12}$$

where 12 comes from Proposition 2. We know that $A(x) \leq 1 = A(x_*)$, that $r(x) \geq cq(x)$, and $\sum_x r(x) = \mathbb{P}[\text{mc} > 0]$. Optimizing subject to these constraints,

$$\sum_x A(x)r(x) < \mathbb{P}[\text{mc} > 0] - \sum_{x \neq x_*} cq(x) + \sum_{x \neq x_*} cq(x)A(x)$$

$$= \mathbb{P}[\text{mc} > 0] + c \sum_x A(x)q(x) - c. \tag{13}$$

So the KL divergence is bounded above by

$$D(p||Ar) < \log\left(c + \frac{\mathbb{P}[\text{mc} > 0] - c}{\sum_x A(x)q(x)}\right) - \log c \tag{14}$$

The quantity $\sum_x A(x)q(x)$ is the probability of accepting a perfect sample from $q(\cdot)$, which Proposition 1 lower bounds:

$$D(p||Ar) < \log\left(c + (1-c)(1+2^{-\gamma})\right) - \log c \tag{15}$$

$$= \log\left(\frac{1}{1+2^{-\Delta}} + \frac{1+2^{-\gamma}}{1+2^{\Delta}}\right) + \log(1+2^{-\Delta}) \tag{16}$$

which for the sake of clarity we can weaken to

$$D(p||Ar) < \log\left(1 + \frac{1+2^{-\gamma}}{1+2^{\Delta}}\right). \tag{17}$$

$\square$

## 2 Accuracy/Runtime trade-off

We analyze the runtime of PROGRAMSAMPLE using number of solver calls as a proxy for runtime. First, we observe that some solver invocations are redundant, as analyzed in Sec. 2.1. Then we give a more thorough overview of how we navigate the trade-off between accuracy and runtime (Sec. 2.2).

### 2.1 Efficient enumeration

The embedding $E$ introduces a symmetry into the solution space of the SAT formula, where one program (an $x$) corresponds to many points in the embedding (pairs $(x, y)$). We more efficiently enumerate surviving members of $E$ by only enumerating unique surviving programs, and then counting the corresponding members of $E$ implicitly through the following result:

**Proposition 4.** *Let $x \in X$ and $(x, y) \in E$ satisfy $h(x, y) \overset{2}{\equiv} b$. If $|x| \geq d$ then $(x, y)$ is the only surviving member of $E$ corresponding to $x$. Otherwise there are $2^{d-|x|-rank(g)}$ survivors where $g$ is the rightmost $d - |x|$ columns of $h$.*

*Proof.* If $|x| \geq d$ then there is only one element of $E$ corresponding to $x$. Otherwise, any assignment to $y$ satisfying

$$b \stackrel{2}{\equiv} \left( \begin{array}{ccccc} h_x & \vdots & h_y & \vdots & g \end{array} \right) \left( \begin{array}{c} x \\ y_{\leq |x|} \\ y_{>|x|} \end{array} \right) \tag{18}$$

satisfies the random hashing constraints, where we have partitioned the columns of $h$ into those multiplied into $x$, $y_{\leq |x|}$, and $y_{>|x|}$. Because $(x, y) \in E$ the values of $x$ and $y_{\leq |x|}$ are fixed, so we can define a new vector $c \stackrel{2}{\equiv} b + h_x x + h_y y_{\leq |x|}$ and rewrite Eq. 18 as $c = g y_{>|x|}$. Let $r = \text{rank}(g)$. Then there is a coordinate system where Eq. 18 reads

$$c \stackrel{2}{\equiv} \left( \begin{array}{cccc} 1 & & & \\ & \ddots & & \\ & & 0 & \\ & & & \ddots \end{array} \right) \left( \begin{array}{c} y_{1+|x|} \\ \vdots \\ y_{1+|x|+r} \\ \vdots \end{array} \right) \tag{19}$$

Eq. 19 is satisfied iff, for all $1 \leq j \leq r$, $y_{j+|x|} = c_j$. For $j > r$ the entries of $y_{j+|x|}$ are unconstrained, and so $2^{d-|x|-r}$ satisfying values for $y$ exist. $\square$

This enumeration strategy helps when sampling from sharply peaked posteriors, where there are few surviving programs; it also bounds the number of solver invocation to $|X|$.

## 2.2 Balancing accuracy and runtime

**Proposition 5.** *The expected number of calls to the solver per sample is bounded above by* $\frac{1+2^\Delta}{(1+2^{-\gamma})^{-1}(1+2^{-\Delta})^{-1}-2^{-\Delta}}$.

*Proof.* First upper bound the probability of failing, $\mathbb{P}[\text{fail}]$, to get a sample, which could happen if $\mathcal{S}$ is empty or if the sample from $\mathcal{S}$ is rejected, which is distributed according to $r(\cdot)$:

$$\mathbb{P}[\text{fail}] < \mathbb{P}[\text{reject}] + \mathbb{P}[\text{mc} = 0], \text{ union bound} \tag{20}$$

$$< 1 - \frac{\sum_x A(x)q(x)}{1 + 2^K/|E|} + 2^K/|E|, \text{ Prop. 2} \tag{21}$$

$$< 1 - \frac{1}{(1 + 2^{-\Delta})(1 + 2^{-\gamma})} + 2^{-\Delta}, \text{ Prop. 1} \tag{22}$$

The expected number of solver invocations per iteration is $< 1 + E[\text{mc}] = 1 + |E|2^{-K} = 1 + 2^\Delta$ and the expected number of iterations is $1/\mathbb{P}[\neg\text{failure}]$. Because the iterations are independent the expected number of solver invocations is just their product, which is the desired result. $\square$

Proposition 5 shows that the number of invocations to the solver grows exponentially in $\Delta$, while Proposition 3 shows that the KL divergence from $p(\cdot)$ decays exponentially in $\Delta$. Algorithm 1 navigates this trade-off through its preliminary model counting steps; see Fig. 1.

Figure 1: Accuracy (colored contours) vs Performance (monochrome cells) trade-off for a program synthesis problem; upper bounds plotted. Performance measured in expected solver invocations; accuracy measured in log KL divergence. Prop. 1 lower bounds the tilt of performant samplers, while Prop. 2 upper bounds $K$ to $O(d)$, forcing our sampler into the darker (faster) regions. KL divergence falls off exponentially fast in $\Delta = O(d - K)$, (Prop. 3) while solver invocations grows exponentially in $\Delta$ (Prop. 5) but is bounded by $|X|$ (Prop. 4), shown in white.

## 3   Comparison to other approaches

We compared with the PAWS [2] variant described in the main paper. Like PROGRAMSAMPLEand other sampling algorithms based on random parity constraints, this algorithm comes with parameters that trade off performance with accuracy. Our baseline is equivalent to setting the $b$ parameter of [2] to 1, which maximally prefers performance over accuracy.

We also attempted a random projection baseline that does not use an embedding. This alternative approach was actually the first we tried, and as far as we know it is unpublished. We now describe this alternative baseline:

Our prior over programs suggests a particularly simple sampling algorithm. The prior is $\mathbb{P}(x) \propto 2^{-|x|}$, where $|x|$ is the number of bits needed to specify the program $x$. So sampling uniformly over assignments to all bits would also sample from our description length prior: let $n$ be the number of Boolean decision variables (bits) that specify the structure of the program. Then the probability of sampling a program $x$ is just $\propto 2^{n-|x|} \propto 2^{-|x|}$. Here we are now assuming that the mapping from Boolean decision variables to programs is many-to-one, which contrasts with PROGRAMSAMPLE, where the mapping is constrained to be one-to-one. Uniform sampling of assignments to these Boolean decision variables can be accomplished with random XOR constraints.

Because this approach avoids the need for any embedding of the program space, one might think that it maybe would sample programs faster in practice. However, the number of random constraints $K$ needed in order to have $o(1)$ survivors is $n + \log Z - \log o(1) \geq n - |x_*| - \log o(1)$. So for very large $n$, which occurs when we consider program spaces that might have long programs, the number of constraints also becomes very large. This explosion in the number of constraints serves to further entangle otherwise independent Boolean decision variables. In practice we found that this baseline causes our solver to timeout after an hour on highly-tilted program induction problems (text edit/counting), to also timeout on our reversing problems (which are intermediate in their tilt), and

to only produce any samples before the timeout on our easiest sorting problem (learning from 5 examples). This pattern of errors is diagnostic of the phenomena we attempt to remedy – namely, the *easiest* program induction problem (counting) became intractable with the naive application of these techniques! Studying these failures led to the development of PROGRAMSAMPLE.

A new solver-aided scheme for weighted model counting is introduced in [3]. Like PAWS or PROGRAMSAMPLE, the main idea is to reduce a weighted counting or sampling problem to an unweighted problem in a higher dimensional space. Although it has not yet been tried as far as we know, the counting scheme in [3] might be adapted to a sampling algorithm which could also efficiently sample from our posteriors over programs.

## 4  Text edit problems

We drew program learning problems from [4] and adapted them to our subset of FlashFill. Below are the problems we tested on. We systematically used either the first one, two, or three input/output examples as training data and the remaining as test data. After each program learning problem we show the program learned from the first three examples, followed by its description length measured in bits.

| Input | Output |
| --- | --- |
| "My name is John." | "John" |
| "My name is Bill." | "Bill" |
| "My name is May." | "May" |
| "My name is Mary." | "Mary" |
| "My name is Josh." | "Josh" |

Program: `SubString(Pos([' '],[],2),-2)`. 20 bits.

| Input | Output |
| --- | --- |
| "james" | "james." |
| "charles" | "charles." |
| "thomas" | "thomas." |
| "paul" | "paul." |
| "chris" | "chris." |

Program: `Append(SubString(0,-1),Const(['.']))`. 22 bits.

| Input | Output |
| --- | --- |
| "don steve g." | "dsg" |
| "Kevin Jason Mat" | "KJM" |
| "Jose Larry S" | "JLS" |
| "Arthur Joe Juan" | "AJJ" |
| "Raymond F. Timothy" | "RFT" |

Program: `Append(Append(SubString(0,1),SubString(Pos([' '],[],0),Pos([' '],[],0))),SubString(Pos([' '],[],3),Pos([' '],[],1)))`. 55 bits.

| Input | Output |
| --- | --- |
| "brent.hard@ho" | "brent hard" |
| "matt.ra@yaho" | "matt ra" |
| "jim.james@har" | "jim james" |
| "ruby.clint@g" | "ruby clint" |
| "josh.smith@g" | "josh smith" |

Program: `Append(Append(SubString(0,Pos([],['.'],3)),Const([' ']))),SubString(Pos(['.'],[],0),Pos([],['@'],0)))`. 57 bits.

| Input | Output |
|---|---|
| "John DOE 3 Data [TS]865-000-0000 - - 453442-00 06-23-2009" | "865-000-0000" |
| "A FF MARILYN 30'S 865-000-0030 - 4535871-00 07-07-2009" | "865-000-0030" |
| "A GEDA-MARY 100MG 865-001-0020 - - 5941-00 06-23-2009" | "865-001-0020" |
| "Sue DME 42 [ST]865-003-0100 – 5555-99 08-22-2010" | "865-003-0100" |
| "Edna DEECS [SSID] 865-001-0003 –23954-11 09-01-2010" | "865-001-0003" |

Program: `Append(Const(['8']),SubString(Pos(['8'],[],0),Pos([],[' ','-'],0)))`. 44 bits.

| Input | Output |
|---|---|
| "Company/Code/index.html" | "Company/Code/" |
| "Company/Docs/Spec/specs.doc" | "Company/Docs/Spec/" |
| "Work/Presentations/talk.ppt" | "Work/Presentations/" |
| "Work/Records/2010/January.dat" | "Work/Records/2010/" |
| "Proj/Numerical/Simulators/NBody/nbody.c" | "Proj/Numerical/Simulators/NBody/" |

Program: `SubString(0,Pos(['/'],[],3))`. 20 bits.

| Input | Output |
|---|---|
| "hi" | "hi hi" |
| "bye" | "hi bye" |
| "adios" | "hi adios" |
| "joe" | "hi joe" |
| "icml" | "hi icml" |

Program: `Append(Const(['h', 'i', ' ']),SubString(0,-1))`. 34 bits.

| Input | Output |
|---|---|
| "Oege de Moor" | "Oege de Moor" |
| "Kathleen Fisher AT" | "T Labs" |
| "Kathleen Fisher AT" | "T Labs" |
| "Microsoft Research" | "Microsoft Research" |
| "John Morse Institute" | "John Morse Institute" |
| "Jennifer Smith Law Firm" | "Jennifer Smith Law Firm" |

Program: `SubString(0,-1)`. 12 bits.

| Input | Output |
|---|---|
| "1/21/2001" | "01" |
| "22.02.2002" | "02" |
| "2003-23-03" | "03" |
| "21/1/2001" | "01" |
| "5/5/1987" | "87" |

Program: `SubString(-3,-1)`. 12 bits.

| Input | Output |
|---|---|
| "Eyal Dechter" | "Dechter, Eyal" |
| "Joshua B. Tenenbaum" | "Tenenbaum, Joshua B." |
| "Stephen H. Muggleton" | "Muggleton, Stephen H." |
| "Kevin Ellis" | "Ellis, Kevin" |
| "Dianhuan Lin" | "Lin, Dianhuan" |

Program: `Append(Append(SubString(Pos([' '],[],3),-1),Const([',', ' '])),SubString(0,Pos([],[' '],3)))`. 55 bits.

| Input | Output |
|---|---|
| "12/31/13" | "12.31" |
| "1/23/2009" | "1.23" |
| "4/12/2023" | "4.12" |
| "6/23/15" | "6.23" |
| "7/15/2015" | "7.15" |

Program:
`Append(Append(SubString(0,Pos([],['/'],0)),Const(['.'])),SubString(Pos(['/'],[],0),Po`
57 bits.

| Input | Output |
|---|---|
| "Three <2: vincent> Jeff" | "(2: vincent)" |
| "Don Kyle <3: ricky> virgil" | "(3: ricky)" |
| "herbert is <2: marion> morris" | "(2: marion)" |
| "fransisco eduardo <1: apple trees>" | "(1: apple trees)" |
| "country music <9: refrigerator>" | "(9: refrigerator)" |

Program:
`Append(Append(Const(['(']),SubString(Pos(['<'],[],3),Pos([],['>'],3))),Const([')']))`.
47 bits.

| Input | Output |
|---|---|
| "3113 Greenfield Ave., Los Angeles, CA 90034" | "Los Angeles" |
| "43 St. Margaret St. 1, Dorchester, MA 02125" | "Dorchester" |
| "43 Vassar St. 46-4053, Cambridge, MA 02139" | "Cambridge" |
| "47 Foskett St. 2, Cambridge, MA 02144" | "Cambridge" |
| "3 Ames St., Portland, OR 02142" | "Portland" |

Program: `SubString(Pos([',', ' '],[],0),Pos([],[','],3))`. 34 bits.

| Input | Output |
|---|---|
| "Verlene Ottley " | "V.O" |
| "Oma Cornelison " | "O.C" |
| "Marin Lorentzen " | "M.L" |
| "Annita Nicely " | "A.N" |
| "Joanie Faas " | "J.F" |

Program: `Append(Append(SubString(0,0),Const(['.'])),SubString(Pos([' '],[],0),Pos([' '],[],0)))`. 43 bits.

| Input | Output |
|---|---|
| "Agripina Kuehner " | "Hi Agripina!" |
| "Brittany Alarcon " | "Hi Brittany!" |
| "Adelia Swindell " | "Hi Adelia!" |
| "Marcie Michalak " | "Hi Marcie!" |
| "Eugena Eurich " | "Hi Eugena!" |

Program: `Append(Append(Const(['H', 'i', ' ']),SubString(0,Pos([],[' '],0))),Const(['!']))`. 51 bits.

| Input | Output |
|---|---|
| "include <stdio.h>" | "stdio" |
| "include <malloc.h>" | "malloc" |
| "include <stdlib.h>" | "stdlib" |
| "include <sys.h>" | "sys" |
| "include <os.h>" | "os" |

Program: `SubString(Pos(['<'],[],3),-4)`. 20 bits.

| Input | Output |
|---|---|
| "aa" | "aaa" |
| "abc" | "abcc" |
| "xyz" | "xyzz" |
| "4" | "44" |
| "john" | "johnn" |

Program: `Append(SubString(0,-1),SubString(-2,-1))`. 24 bits.

| Input | Output |
|---|---|
| "3113 Greenfield Ave., LA, CA 90034" | "3113" |
| "43 St. Margaret St. 1, Dorchester, MA 02125" | "43" |
| "43 Vassar St. 46-4053, Cambridge, MA 02139" | "43" |
| "47 Foskett St. 2, Cambridge, MA 02144" | "47" |
| "3 Ames St., Portland, OR 02142" | "3" |

Program: `SubString(0,Pos([],[' '],0))`. 20 bits.

| Input | Output |
|---|---|
| "aa" | "aaaa" |
| "abc" | "abcabc" |
| "xyz" | "xyzxyz" |
| "4" | "44" |
| "john" | "johnjohn" |

Program: `Append(SubString(0,-1),SubString(0,-1))`. 24 bits.