[Reviews · NeurIPS 2016]

Reviewer 1

Summary

This paper describes an algorithm for sampling from the set of programs consistent with a given set of input-output training examples. The programs come from a finite set of possible programs defined by a "sketch" (which seems to be essentially a grammar), with a description length prior. The paper builds on the existing approach of program synthesis by constraint solving, where the possible programs are encoded as assignments to a set of Boolean variables, and a SAT solver is used to find programs consistent with the examples. The advance in this paper is that rather than just finding a single most likely solution, the algorithm samples from the posterior. The paper provides an error bound on the sampling probabilities that allows trading off time versus accuracy.

Qualitative Assessment

I found this paper interesting and well-written, but I have some significant questions and comments about the approach. * Figure 5 (page 6) shows that the existing MDL approach of finding a single most likely program does slightly better than the proposed sampling approach in terms of average accuracy. The paper argues that sampling is useful because we can find the C most frequently sampled programs and show them to a user. As shown in Figure 6, there is more likely to be a correct program in the top 3 programs than in the top 1. But if we want to show the top C programs, do we really need to perform sampling, which the paper says is complicated by the existence of many long and unlikely programs that match the training examples? Why can't we simply find the MDL program and then run the solver again with length restrictions to find other consistent programs of the same length, or slightly longer lengths? * Page 5, line 179 says that the paper "implicitly defined a description-length prior, where |x| is the number of bits required to specify x in the SAT encoding". Looking at the constraints in Figure 2(c), it appears that if bits x_1, ..., x_k fully define a program, then bits x_{k+1}, ..., x_n end up unconstrained. So is a program of length k represented by 2^{n-k} distinct assignments to the bits? Is that what's meant by an implicitly defined prior? If so, then it seems that the prior is uniform over bit assignments (although not over programs), and I don't see the point of mapping from a uniform distribution on a separate embedding space as discussed in Section 2. * For string manipulation, the paper uses data sets of size 5, with 1-3 examples used for training. This leaves only 2-4 examples for testing, which doesn't seem like enough to yield reliable results. For list manipulation, I couldn't find a statement about the size of the test sets. I assume correct programs for these example problems are known, so it should be straightforward to generate tests sets of at least size 100. Using larger test sets would reduce the chance that some of the returned programs yield high test accuracy by accident. == Update After Author Feedback == I appreciate the authors' responses to my comments. I think I did misinterpret Figure 6; I understand now that it is referring to the top C most frequent combinations of program outputs, not the top C most frequent programs. Still, it would be informative to compare explicitly to a K-best MDL approach, to demonstrate that integrating out the program really helps. I'm still wondering about how the prior distribution over programs ends up being defined. I asked in my review if a program of length k ended up being represented by 2^{n-k} distinct assignments to the bits in the SAT encoding, and the response seemed to agree that this was true. But then why do we need to add auxiliary variables in the embedding space as described on page 4? Isn't it enough to sampling from a uniform distribution over bit assignments? If we start with a space where shorter programs already account for more assignments, and then we also add the auxiliary variables, don't we end up with a *squared* MDL prior? I hope this can be clarified in the final version.

Confidence in this Review

2-Confident (read it all; understood it all reasonably well)


Reviewer 2

Summary

Bayesian program learning is an important problem that considers the uncertainty or ambiguity, especially when the number of training examples is small. However, drawing samples from a Bayesian posterior distribution is often challenging due to the combinatorially large sample space and the high-tilted distribution. By assuming that the candidate programs are a subset of those handled by constraint-based program synthesis tools, this paper presents a sampling algorithm under this regime. The method first builds a low-tilt approximation distribution, whose discrepancy is corrected by rejection sampling, and then adopts a variant of the embedding-and-projection scheme [8] to draw samples. The approximation error is analyzed in proposition 3. The empirical performance was evaluated on two program learning problems in text editing and list/manipulation domains.

Qualitative Assessment

Overall, this paper is well-written. It tries to address an important problem under a reasonable assumption (or restriction). The proposed method was analyzed both theoretically and empirically. However, some key parameters are not clear from the main text. For example, what value of d (and |x_*|) was used exactly in the experiments? How sensitive is it w.r.t d? The paper claimed that previous methods can be very low efficiency, but without reporting explicitly how efficient they are as compared to the proposed methods (e.g., Table 1 only reports the time of the proposed method, no mention on others). A final tiny point: what do you mean by 20 in line 89?

Confidence in this Review

2-Confident (read it all; understood it all reasonably well)


Reviewer 3

Summary

The paper proposed a new method that extends previous techniques to sample programs with the extremely tilted distribution.

Qualitative Assessment

1.My general impression is whether the case, programs extremely tilted distribution, is common. 2.The proposed method has a bias towards shorter or simpler programs, why? 3.The proposed method twice distorts the posterior distribution. However, previous techniques just need one distortion. Does the proposed method leads to higher computational complexity?

Confidence in this Review

2-Confident (read it all; understood it all reasonably well)


Reviewer 4

Summary

The authors propose an algorithm that uses a symbolic solver to efficiently sample programs under a Bayesian learning paradigm. The proposed approach combines program synthesis (constraint-based) and sampling (via parity constraints). The authors provide theoretical guarantees and present experimental results (text editing and list manipulation) highlighting the efficiency and performance characteristics of the proposed approach.

Qualitative Assessment

The methodology presented in the paper seems sound and the experimental results are extensive and convincing, however I could not evaluate the theoretical contributions as well as in the impact of the paper in context with related existing work because some of the technical elements presented in the paper are beyond my area of expertise.

Confidence in this Review

1-Less confident (might not have understood significant parts)


Reviewer 5

Summary

The paper presents a method (called ProgramSample) for sampling from the posterior distribution of programs which satisfy some constraints specified as input/output examples. The prior is a minimum description length criterion, and the space of possible programs is delimited by a sketch (i.e. generating grammar). Directly drawing posterior samples from this space is challenging because the posterior distribution often has high tilt, very few likely (short) programs and many unlikely (long) programs. To combat this, the proposed method (1) uses a bounded-tilt approximation q to the posterior p, (2) defines a higher-dimensional embedding space in which uniform sampling corresponds to sampling from q, and (3) uses random projections to efficiently uniformly sample a distribution r which can be made arbitrarily close to q. The authors evaluate the method on text editing (i.e. string manipulation) programs as well as list manipulation programs, demonstrating that sampled programs can generalize to new unseen example, and that integrating out the program itself can in some cases lead to better performance than using the MAP program only.

Qualitative Assessment

I found this paper to be extremely well-written, the method well-motivated, the explanations clear, and the evaluations sufficient. I especially commend the authors for the thorough explanation of limitations in Section 4 (a subject that often does not receive as much attention as it should). My questions and comments concern mostly minor issues: Figure 5: the meaning of the axes aren't clear to me. They should be defined in either the figure caption or the main text. Figure 6: Perhaps I've just missed something, but why don't we see the benefits of sampling in the Figure 5 results, as well? The caption text suggests that perhaps this is because the figure depicts a one-shot learning scenario, but then again, Figure 5 also has curves for "1 example"... On a related note: what problem does Figure 6 present results for? The main text suggests that it might be the ambiguous problem from Figure 1, but this should be made explicit. Can you give (perhaps in Table 1) data on how large n (and d) are for the problems you considered? It would help to build intuition for how compactly (or not) these programs can be encoded as bitvectors, and how that affects runtime. This is alluded to in the limitations section, but it would be nice to see some numbers. Typos: - Figure 5 legend: "1 examples" - Line 224: "Systems with this goal has been successful..." has -> have

Confidence in this Review

2-Confident (read it all; understood it all reasonably well)


Reviewer 6

Summary

The paper is about a framework called PROGRAMSAMPLE, which quickly learns a program given only a few examples. The approach samples from an hypothesis space, called sketch, which is necessary to indicate in before. There exists other approaches as for example MCMC, which are more high level than the presented approach, but could get trapped in local minima.

Qualitative Assessment

For me, a non professional person in this area, this paper is not understandable. - Please introduce: - Bayesian Program Learning. - description-length prior - SMT solver (what is the abbreviation SMT for?) - SAT? - MAP (299)? ...please introduce all abbreviations, also AI (Line 266)! - Please sort your Figures in a way they appear in the text (Figure 4 is under Figure 5 and 6) (Figure 7 is referenced in the text below Figure 8 and 9) - Please explain Figure 8. It is not clear, what you mean with the single statements. - In Figure 2, you neither explain the symbol P_i, i=1,...n in the Figure caption or in the text.

Confidence in this Review

1-Less confident (might not have understood significant parts)